# LEARNING SEMANTIC WORD RESPRESENTATIONS VIA TENSOR FACTORIZATION

## ABSTRACT

Many state-of-the-art word embedding techniques involve factorization of a co-occurrence based matrix. We aim to extend this approach by studying word embedding techniques that involve factorization of co-occurrence based *tensors* ($N$-way arrays). We present two new word embedding techniques based on tensor factorization, and show that they outperform multiple commonly used embedding methods when used for various NLP tasks on the same training data. Also, to train one of the embeddings, we present a new joint tensor factorization problem and an approach for solving it. Furthermore, we modify the performance metrics for the Outlier Detection task (Camacho-Collados & Navigli (2016)) to measure the quality of higher-order relationships that a word embedding captures. Our tensor-based methods significantly outperform existing methods at this task when using our new metric. Finally, we demonstrate that vectors in our embeddings can be composed *multiplicatively* to create different vector representations for each meaning of a polysemous word in a way that cannot be done with other common embeddings. We show that this property stems from the higher order information that the vectors contain, and thus is unique to our tensor based embeddings.

## INTRODUCTION

Word embeddings have been used to improve the performance of many NLP tasks including language modelling (Bengio et al. (2003)), machine translation (Bahdanau et al. (2014)), and sentiment analysis (Kim (2014)). The broad applicability of word embeddings to NLP implies that improvements to their quality will likely have widespread benefits for the field.

The word embedding problem is to learn a mapping $\eta : V \to \mathbb{R}^k$ ($k \approx 100$-300 in most applications) that encodes meaningful semantic and/or syntactic information. For instance, in many word embeddings, $\eta(\text{car}) \approx \eta(\text{truck})$, since the words are semantically similar.

More complex relationships than similarity can also be encoded in word embeddings. For example, we can answer analogy queries of the form $a : b :: c : ?$ using simple arithmetic in many state-of-the-art embeddings (Mikolov et al. (2013)). The answer to bed : sleep :: chair : $x$ is given by the word whose vector representation is closest to $\eta(\text{sleep}) - \eta(\text{bed}) + \eta(\text{chair})$ ($\approx \eta(\text{sit})$). Other embeddings may encode such information in a nonlinear way (Jastrzebski et al. (2017)).

Mikolov et al. (2013) demonstrates the additive compositionality of their `word2vec` vectors: one can sum vectors produced by their embedding to compute vectors for certain *phrases* rather than just vectors for words. Later in this paper, we will show that our embeddings naturally give rise to a form of *multiplicative* compositionality that has not yet been explored in the literature.

Almost all recent word embeddings rely on the distributional hypothesis (Harris), which states that a word's meaning can be inferred from the words that tend to surround it. To utilize the distributional hypothesis, many embeddings are given by a low-rank factor of a matrix derived from co-occurrences in a large unsupervised corpus. For examples of this, see Pennington et al. (2014); Murphy et al. (2012); Levy & Goldberg (2014) and Salle et al. (2016). Approaches that rely on matrix factorization only utilize pairwise co-occurrence information in the corpus. We aim to generalize this approach by creating word embeddings given by factors of *tensors* containing higher order co-occurrence data.

## RELATED WORK

Some common word embeddings related to co-occurrence based matrix factorization include GloVe (Pennington et al. (2014)), `word2vec` (Levy & Goldberg (2014)), LexVec (Salle et al. (2016)), and NNSE (Murphy et al. (2012)). In contrast, our work studies word embeddings given by factorization of tensors. An overview of tensor factorization methods is given in Kolda & Bader (2009).

In particular, our work uses symmetric factorization of symmetric tensors. Theoretical foundations for this problem have been studied and outlined in (Comon et al. (2008)). We note that while finding the best rank-$R$ nonnegative approximation of nonnegative tensors is a well-posed problem (Lim & Comon (2009a), Qi et al. (2016)), relatively little is known about guarantees on the existence of best rank-$R$ symmetric tensor approximation of a symmetric tensor, which may not exist in general, especially over $\mathbb{R}$ (Qi et al. (2017))[1]. Further, it may be NP-hard to obtain these factorizations in general. Nevertheless, as is done in many applications that factor the third order cumulant, we are merely seeking a (not necessarily best) real symmetric rank-$R$ approximation to the symmetric tensor.

Recently, under incoherency assumptions on the factors, a method using factorization of symmetric tensors was proposed to create a generic word embedding Sharan & Valiant (2017), but the results were not evaluated extensively. Our work studies this idea in much greater detail, fully demonstrating the viability of tensor factorization as a technique for training word embeddings.[2] In general, tensor factorization has also been applied to NLP (Van de Cruys et al. (2013); Zhang et al. (2014)), as has nonnegative factorization of nonnegative tensors (Van de Cruys (2009)).

Composition of word vectors to create novel representations has been studied in depth, including additive, multiplicative, and tensor-based methods (Mitchell & Lapata (2010); Blacoe & Lapata (2012)). Typically, composition is used to create vectors that represent *phrases* or *sentences*. Our work, instead, shows that pairs of word vectors can be composed multiplicatively to create different vector representations for the various meanings of a *single* polysemous word.

Finally, we note that although other tensor factorizations such as the Higher-Order SVD (Kolda & Bader (2009)) and Tensor Train (TT) (Oseledets (2011)) may be adapted to find word embeddings from the co-occurance tensor, in this work we only consider the symmetric CP decomposition, leaving the study of other tensor decompositions to future work.

## MATHEMATICAL PRELIMINARIES

### NOTATION

Throughout this paper we will write scalars in lowercase italics $\alpha$, vectors in lowercase bold letters $\mathbf{v}$, matrices with uppercase bold letters $\mathbf{M}$, and tensors (of order $N > 2$) with Euler script notation $\mathcal{X}$, as is standard in the literature.

### POINTWISE MUTUAL INFORMATION

Pointwise mutual information (PMI) is a useful property in NLP that quantifies the likelihood that two words co-occur (Levy & Goldberg (2014)). It is defined as:

$$PMI(x, y) = \log \frac{p(x, y)}{p(x)p(y)}$$

where $p(x, y)$ is the probability that $x$ and $y$ occur together in a given fixed-length context window in the corpus, irrespective of order.

It is often useful to consider the *positive* PMI (PPMI), defined as:

$$PPMI(x, y) := \max(0, PMI(x, y))$$

---

[1]See also - https://www.stat.uchicago.edu/ lekheng/work/msri-lect2.pdf - slide 31

[2]In this work we are don't consider Orthogonal-ALS, proposed in Sharan & Valiant (2017), and leave its extensive evaluation for future work.

since negative PMI values have little grounded interpretation (Bullinaria & Levy (2007); Levy & Goldberg (2014); Van de Cruys (2009)).

Given an indexed vocabulary $V = \{w_1, \ldots, w_{|V|}\}$, one can construct a $|V| \times |V|$ PPMI matrix $\mathbf{M}$ where $m_{ij} = PPMI(w_i, w_j)$. Many existing word embedding techniques involve factorizing this PPMI matrix: Levy & Goldberg (2014); Murphy et al. (2012); Salle et al. (2016).

PMI can be generalized to $N$ variables. While there are many ways to do so (Van de Cruys (2011)), in this paper we use the form defined by:

$$PMI(x_1^N) = \log \frac{p(x_1, \ldots, x_N)}{p(x_1) \cdots p(x_N)}$$

where $p(x_1, \ldots, x_N)$ is the probability that *all* of $x_1, \ldots, x_N$ occur together in a given fixed-length context window in the corpus, irrespective of their order.

In this paper we study 3-way PPMI *tensors* $\mathbf{M}$, where $m_{ijk} = PPMI(w_i, w_j, w_k)$, as this is the natural higher-order generalization of the PPMI matrix. We leave the study of creating word embeddings with $N$-dimensional PPMI tensors ($N > 3$) to future work.

TENSOR FACTORIZATION

Just as the rank-$R$ matrix decomposition is defined to be the product of two factor matrices ($\mathbf{M} \approx \mathbf{U}\mathbf{V}^\top$), the canonical rank-$R$ tensor decomposition for a third order tensor is defined to be the product of three factor matrices (Kolda & Bader (2009)):

$$\mathbf{X} \approx \sum_{r=1}^{R} \mathbf{u}_r \otimes \mathbf{v}_r \otimes \mathbf{w}_r =: [\![\mathbf{U}, \mathbf{V}, \mathbf{W}]\!], \tag{1}$$

where $\otimes$ is the outer product: $(\mathbf{a} \otimes \mathbf{b} \otimes \mathbf{c})_{ijk} = a_i b_j c_k$. This is also commonly referred to as the rank-$R$ *CP Decomposition*. Elementwise, this is written as:

$$x_{ijk} \approx \sum_{r=1}^{R} u_{ir} v_{jr} w_{kr} = \langle \mathbf{u}_{:,i} * \mathbf{v}_{:,j}, \mathbf{w}_{:,k} \rangle,$$

where $*$ is elementwise vector multiplication and $\mathbf{u}_{:,i}$ is the $i^{th}$ row of $\mathbf{U}$. In our later section on multiplicative compositionality, we will see this formulation gives rise to a meaningful interpretation of the elementwise product between vectors in our word embeddings.

**Symmetric CP Decomposition.** In this paper, we will consider *symmetric* CP decomposition of *nonnegative* tensors (Lim (2005); Kolda & Bader (2009)). Since our $N$-way PPMI is nonnegative and invariant under permutation, the PPMI tensor $\mathbf{M}$ is nonnegative and supersymmetric, i.e. $m_{ijk} = m_{\sigma(i)\sigma(j)\sigma(k)} \geq 0$ for any permutation $\sigma \in S_3$.

In the symmetric CP decomposition, instead of factorizing $\mathbf{M} \approx [\![\mathbf{U}, \mathbf{V}, \mathbf{W}]\!]$, we factorize $\mathbf{M}$ as the triple product of a *single* factor matrix $\mathbf{U} \in \mathbb{R}^{|V| \times R}$ such that

$$\mathbf{M} \approx [\![\mathbf{U}, \mathbf{U}, \mathbf{U}]\!]$$

In this formulation, we use $\mathbf{U}$ to be the word embedding so the vector for $w_i$ is the $i^{th}$ row of $\mathbf{U}$ similar to the formulations in: Levy & Goldberg (2014); Murphy et al. (2012); Pennington et al. (2014).

WHY FACTORIZE THE THIRD MOMENT?

Factorizing the PPMI tensor of the third moment of co-occurrence is a natural extension of current methods and justifiable for a number of other reasons. For one, if NLP tasks such as the semantic analogy task depend on how embedding vectors cluster, the goal of training a word embedding is to find a map of words to vectors such that the vectors for semantically similar words form a cluster. For identifying the clusters of a planted partition model such as the Stochastic Block Model (SBM), the spectral factorization of node interactions completely derived from pair-wise interactions

is sufficient for discovering the disjoint clusters (Belkin & Niyogi (2001); Krzakala et al. (2013); Spielman (2007)).

The existence of polysemous words in the corpus necessitate the assumption of a Mixed Membership (MM) model, since polysemous words belong to multiple different clusters of meaning (Foulds (2017)). In this case, it is well-known that factorizing the third moment provably recovers the parameters of planted Mixed Membership-SBM model (Anandkumar et al. (2013)) in a way that only capturing pair-wise interactions, i.e. the second order moment (uniquely) cannot. Further, from the perspective of using Gaussian mixture models for capturing polysemy in word embeddings (Athiwaratkun & Wilson (2017)) it is known that factorizing the third moments can provably identify the isotropic Gaussian mixture models (Anandkumar et al. (2014)).

For an analysis of the degree to which our tensor factorization-based embeddings capture polysemy, we refer the reader to Section on multiplicative compositionality on Page 9. Another perspective is that considering the third order moment further contextualizes the co-occurrence matrix, adding information that was lost by only considering matrix factorization.

While in this work we do not require any provable results for the derived word embeddings, we are motivated by several recent developments successfully applying tensor factorization for addressing related machine learning problems.

## METHODOLOGIES

### COMPUTING THE SYMMETRIC CP DECOMPOSITION

The $\Theta(|V|^3)$ size of the third order PPMI tensor presents a number of computational challenges. In practice, $|V|$ can vary from $10^4$ to $10^6$, resulting in a tensor whose naive representation requires at least $4 * 10,000^3$ bytes $= 4$ TB of floats. Even the sparse representation of the tensor takes up such a large fraction of memory that standard algorithms such as successive rank-1 approximation (Wang & Qi (2007); Mu et al. (2015)) and alternating least-squares (Kolda & Bader (2009)) are infeasible for our uses. Thus, in this paper we will consider a stochastic online formulation similar to that of (Maehara et al. (2016)).

We optimize the CP decomposition in an online fashion, using small random subsets $\mathcal{M}^t$ of the nonzero tensor entries to update the decomposition at time $t$. In this minibatch setting, we optimize the decomposition based on the current minibatch and the previous decomposition at time $t - 1$. To update $\mathbf{U}$ (and thus the symmetric decomposition), we first define an $n$-dimensional decomposition loss $\mathcal{L}^{(n)}(\mathcal{M}^t, \mathbf{U})$ and minimize this loss with respect to $\mathbf{U}$ using Adam (Kingma & Ba (2014)).

At each time $t$, we take $\mathcal{M}^t$ to be all co-occurrence triples (weighted by PPMI) in a fixed number of sentences (around 1,000) from the corpus. We continue training until we have depleted the entire corpus.

For $\mathcal{M}^t$ to accurately model $\mathcal{M}$, we also include a certain proportion of elements with zero PPMI (or "negative samples") in $\mathcal{M}^t$, similar to that of Salle et al. (2016).

We use empirically-found hyperparameters for training using Random Search (Bergstra & Bengio (2012)) and leave theoretical discovery of optimal hyperparameters (such as negative sample proportion) to future work.

### WORD EMBEDDING PROPOSALS

**CP-S.** The first embedding we propose is based on symmetric CP decomposition of the PPMI tensor $\mathcal{M}$ as discussed in the mathematical preliminaries section. The optimal setting for the word embedding $\mathbf{W}$ is:

$$\mathbf{W} := \operatorname*{argmin}_{\mathbf{U}} ||\mathcal{M} - [\![\mathbf{U}, \mathbf{U}, \mathbf{U}]\!]||_F$$

Since we cannot feasibly compute this exactly, we minimize the loss function defined as the squared error between the values in $\mathcal{M}^t$ and their predicted values:

$$\mathcal{L}^{(3)}(\mathcal{M}^t, \mathbf{U}) = \sum_{m_{ijk}^t \in \mathcal{M}^t} (m_{ijk}^t - \sum_{r=1}^{R} u_{ir} u_{jr} u_{kr})^2$$

using the techniques discussed in the previous section. This idea can be extended to $n$-dimensional tensors by considering the suitable generalization of this loss function given by $\mathcal{L}^{(n)}$, but we only consider at most third order tensors in this work.

**JCP-S.** A potential problem with CP-S is that it is *only* trained on third order information. To rectify this issue, we propose a novel joint tensor factorization problem we call *Joint Symmetric Rank-$R$ CP Decomposition*. In this problem, the input is the fixed rank $R$ and a list of supersymmetric tensors $\mathcal{M}_n$ of different orders but whose axis lengths all equal $|V|$. Each tensor $\mathcal{M}_n$ is to be factorized via rank-$R$ symmetric CP decomposition using a *single* $|V| \times R$ factor matrix $\mathbf{U}$.

To produce a solution, we first define the loss at time $t$ to be the sum of the reconstruction losses of each different tensor:

$$\mathcal{L}_{\text{joint}}((\mathcal{M}^t)_{n=2}^N, \mathbf{U}) = \sum_{n=2}^{N} \mathcal{L}^{(n)}(\mathcal{M}_n^t, \mathbf{U}),$$

where $\mathcal{M}_n$ is an $n$-dimensional supersymmetric PPMI tensor. We then minimize the loss with respect to $\mathbf{U}$. Since we are using at most third order tensors in this work, we assign our word embedding $\mathbf{W}$ to be:

$$\mathbf{W} := \underset{\mathbf{U}}{\text{argmin}} \, \mathcal{L}_{\text{joint}}(\mathbf{M}_2, \mathcal{M}_3, \mathbf{U})$$

$$= \underset{\mathbf{U}}{\text{argmin}} \left[ \mathcal{L}^{(2)}(\mathbf{M}_2, \mathbf{U}) + \mathcal{L}^{(3)}(\mathcal{M}_3, \mathbf{U}) \right]$$

This problem is a specific instance of Coupled Tensor Decomposition which has been studied in the past (Acar et al. (2011); Naskovska & Haardt (2016)). In this problem, the goal is to factorize multiple tensors using *at least* one factor matrix in common. A similar formulation to our problem can be found in Comon et al. (2015), which studies blind source separation using the algebraic geometric aspects of jointly factorizing numerous supersymmetric tensors (to unknown rank). However, we attack the problem numerically while they outline some generic rank properties of such a decomposition rather. In our formulation the rank is fixed and an approximate solution must be found. Exploring the connection between the theoretical aspects of joint decomposition and quality of word embeddings would be an interesting avenue for future work.

To the best of our knowledge this is the first study of Joint *Symmetric Rank-$R$* CP Decomposition, and the first application of Coupled Tensor Decomposition to word embedding.

### SHIFTED PMI

In the same way Levy & Goldberg (2014) considers factorization of positive shifted PMI matrices, we consider factorization of positive shifted PMI *tensors* $\mathcal{M}$, where $m_{ijk} = \max(PMI(w_i, w_j, w_k) - \alpha, 0)$ for some constant shift $\alpha$. We empirically found that different levels of shifting resulted in different qualities of word embeddings – the best shift we found for CP-S was a shift of $\alpha \approx 2.7$, whereas any nonzero shift for JCP-S resulted in a worse embedding across the board. When we discuss evaluation we report the results given by factorization of the PPMI tensors shifted by the best value we found for each specific embedding.

### COMPUTATIONAL NOTES

When considering going from two dimensions to three, it is perhaps necessary to discuss the computational issues in such a problem size increase. However, it should be noted that the creation of pre-trained embeddings can be seen as a pre-processing step for many future NLP tasks, so if the training can be completed once, it can be used forever thereafter without having to take training time into account. Despite this, we found that the training of our embeddings was not considerably

**Table 1:** Best tensor factorization hyperparameter values found

| (Hyperparam) | Tensor value shift | Negative sample percent |
|---|---|---|
| **CP-S** | $-3.9$ | 0.25 |
| **JCP-S (second order)** | 0 | 0.15 |
| **JCP-S (third order)** | $-3.9$ | 0.15 |

slower than the training of order-2 equivalents such as SGNS. Explicitly, our GPU trained CBOW vectors (using the experimental settings found below) in 3568 seconds, whereas training CP-S and JCP-S took 6786 and 8686 seconds respectively.

Also, we are aware of the well-posedness of the nonnegative CP decomposition (Lim & Comon (2009b); Qi et al. (2014)), but found that considering this nonnegative decomposition did not significantly change the results we got on downstream tasks either for CP decomposition or Joint CP decomposition. We thus decide to only report results on the simpler case of unrestricted CP decomposition.

## EVALUATION

In this section we present a quantitative and qualitative evaluation of our embeddings against an informationless embedding and state-of-the-art baselines. Our baselines are:

1. **Random** (random vectors with I.I.D. entries normally distributed with mean 0 and variance $\frac{1}{2}$), for comparing against a model with no meaningful information encoded. This provides a hard baseline that all embeddings should outperform.

2. **Skip-gram with Negative Sampling** (SGNS - Mikolov et al. (2013)), for comparison against the most visible and commonly used embedding method and for comparison against a technique related to PPMI matrix factorization (Levy & Goldberg (2014))

3. **Continuous BOW with Negative Sampling** (CBOW - Mikolov et al. (2013)), for the same reasons as SGNS

4. **Nonnegative Sparse Embedding**[3] (NNSE - Murphy et al. (2012)), for comparison against a technique that also directly uses explicit PPMI matrix factorization

5. **Global Vectors** (GloVe - Pennington et al. (2014)), for comparison against another very commonly used state-of-the art embedding method as well as another matrix factorization-based embedding

For a fair comparison, we trained each model on the same corpus of 10 million sentences gathered from Wikipedia. We removed stopwords and words appearing fewer than 1,000 times (110 million tokens total) to reduce noise and uninformative words.

The baselines were trained using the recommended hyperparameters from their original publications, and all stochastic optimizers were used their default settings. Hyperparameters are always consistent across evaluations. The best hyperparameter settings we found for each embedding are found in Table 1.

Because of the dataset size, the results shown should be considered a proof of concept rather than an objective comparison to state-of-the-art pre-trained embeddings. Due to the natural computational challenges arising from working with tensors, we leave creation of a full-scale production ready embedding based on tensor factorization to future work.

As is commonly done in the literature (Mikolov et al. (2013); Murphy et al. (2012)), we use 300-dimensional vectors for our embeddings and all word vectors are normalized to unit length prior to evaluation.

The numbers presented for all stochastic tasks (such as the supervised ML tasks) are the mean result from 10 random restarts.

---

[3]The input to NNSE is an $m \times n$ matrix, where there are $m$ words and $n$ co-occurrence patterns. In our experiments, we set $m = n = |V|$ and set the co-occurrence information to be the number of times $w_i$ appears within a window of 5 words of $w_j$. As stated in the paper, the matrix entries are weighted by PPMI.

QUANTITATIVE TASKS

**Outlier Detection.** The Outlier Detection task (Camacho-Collados & Navigli (2016)) is to determine which word in a list $L$ of $n+1$ words is unrelated to the other $n$ which were chosen to be related. For each $w \in L$, one can compute its compactness score $c(w)$, which is the compactness of $L \setminus \{w\}$. $c(w)$ is explicitly computed as the mean similarity of all word *pairs* $(w_i, w_j) : w_i, w_j \in L \setminus \{w\}$. The predicted outlier is $\text{argmax}_{w \in L} c(w)$, as the $n$ related words should form a compact cluster with high mean similarity.

We use the WikiSem500 dataset (Blair et al. (2016)) which includes sets of $n = 8$ words per group gathered based on semantic similarity. Thus, performance on this task is correlated with the amount of semantic information encoded in a word embedding. Performance on this dataset was shown to be well-correlated with performance at the common NLP task of sentiment analysis (Blair et al. (2016)).

The two metrics associated with this task are accuracy and Outlier Position Percentage (OPP). Accuracy is the fraction of cases in which the true outlier correctly had the highest compactness score. OPP measures how *close* the true outlier was to having the highest compactness score, rewarding embeddings more for predicting the outlier to be in $2^{nd}$ place rather than $n^{th}$ when sorting the words by their compactness score $c(w)$.

**3-way Outlier Detection.** As our tensor-based embeddings encode higher order relationships between words, we introduce a new way to compute $c(w)$ based on *groups of 3* words rather than pairs of words. We define the compactness score for a word $w$ to be:

$$c(w) = \sum_{\mathbf{v}_{i_1} \neq \mathbf{v}_w} \sum_{\mathbf{v}_{i_2} \neq \mathbf{v}_w, \mathbf{v}_{i_1}} \sum_{\mathbf{v}_{i_3} \neq \mathbf{v}_w, \mathbf{v}_{i_1}, \mathbf{v}_{i_2}} sim(\mathbf{v}_{i_1}, \mathbf{v}_{i_2}, \mathbf{v}_{i_3}),$$

where $sim(\cdot)$ denotes similarity between a group of 3 vectors. $sim(\cdot)$ is defined as:

$$sim(\mathbf{v}_1, \mathbf{v}_2, \mathbf{v}_3) = \left( \frac{1}{3} \sum_{i=1}^{3} ||\mathbf{v}_i - \frac{1}{3} \sum_{j=1}^{3} \mathbf{v}_j||_2 \right)^{-1}$$

We call this evaluation method OD3.

The purpose of OD3 is to evaluate the extent to which an embedding captures $3^{rd}$ order relationships between words. As we will see in the results of our quantitative experiments, our tensor methods outperform the baselines on OD3, which validates our approach.

This approach can easily be generalized to OD$N$ $(N > 3)$, but again we leave the study of higher order relationships to future work.

**Simple supervised tasks.** Jastrzebski et al. (2017) points out that the primary application of word embeddings is *transfer learning* to actual NLP tasks. They argue that to evaluate an embedding's ability to transfer information to a relevant task, one must measure the embedding's *accessibility of information* for actual downstream tasks. To do so, one must observe the performance of simple supervised tasks as training set size increases, which is commonly done in transfer learning evaluation (Jastrzebski et al. (2017)). If an algorithm using a word embedding performs well with just a small amount of training data, then the information encoded in that embedding is easily accessible.

The simple supervised downstream tasks we use to evaluate the embeddings are as follows:

1. **Word classification.** We consider the task of labelling a word's part of speech based solely on its word vector using a simple supervised classifier (Logistic Regression) as suggested in Jastrzebski et al. (2017). Performance at this task is a direct measure of the amount of syntactic information encoded in the embedding. Using different amounts of training data for the classifier will tell us how easily accessible such information is in that embedding. The dataset we use is the Penn Treebank (Marcus et al. (1994)), which contains 35,000 PoS tagged words, and we use a 85/15 train/test split.

2. **Sentiment analysis.** We also consider a sentiment analysis task as described by Schnabel et al. (2015). To train the classifier we use the suggested Large Movie Review dataset (Maas et al. (2011)) which contains 50,000 sentiment-labeled movie reviews.

All code is implemented using scikit-learn or TensorFlow and uses the suggested train/test split.

**Word similarity.** To present results on the most standard word embedding evaluation technique, we evaluate the embeddings using word similarity on the common MEN, MTurk, RW, and SimLex999 datasets (Bruni et al. (2014); Radinsky et al. (2011); Luong et al. (2013); Hill et al. (2015)). For an overview of word similarity evaluation, see Schnabel et al. (2015).

QUANTITATIVE RESULTS

**Table 2:** Outlier Detection scores across all embeddings

| (Method) | OD2 OPP | OD2 acc | OD3 OPP | OD3 acc |
|---|---|---|---|---|
| **Random** | 0.5828 | 0.2504 | 0.5076 | 0.1823 |
| **SGNS** | 0.6219 | 0.3483 | 0.6109 | 0.32 |
| **CBOW** | 0.6012 | 0.3178 | 0.6014 | 0.3014 |
| **NNSE** | 0.6603 | 0.3467 | 0.5486 | 0.2214 |
| **GloVe** | 0.6550 | 0.3500 | 0.5990 | 0.2456 |
| **CP-S** | 0.6671 | 0.3628 | **0.6738** | **0.3358** |
| **JCP-S** | **0.7069** | **0.4181** | 0.6606 | 0.3247 |

**Outlier Detection results.** The results are shown in **Table 2**. As we can see, the tensor factorization based methods outperform the other non-tensor based baselines for all the formulations of this task. CP-S performs the best for OD3 by a decent margin, which is sensible given that is is trained on third order information only. This is also indicative of OD3's ability to measure the amount of third order information encoded in an embedding. Since the WikiSem500 dataset is focused on semantic relationships, performance at this task demonstrates the quality of semantic information encoded in our embeddings.

It is perhaps surprising that CP-S performs so well in the second order case of OD2, but this could be justified by the fact that third order information sheds light on second order information in a way that is not invertible. To illustrate this point, consider three words $w, x, y$. If $w$ often co-occurs with both $x$ and $y$, it is clear that $w$ also often co-occurs with $x$, but the separate statements that $w$ co-occurs with $x$ and $w$ co-occurs with $y$ does not imply anything about the 3-way co-occurrence of $(w, x, y)$. Further, the pairwise co-occurrence of $x$ and $y$ can be extracted from the 3-way co-occurrence data: $\#(x, y) = \sum_{z \in V} \#(x, y, z)$. It is thus believable that utilizing third order co-occurrence can lead to improved performance on tasks that rely on pairwise information.

**Word classification results.** The results are shown in **Figure 1**.

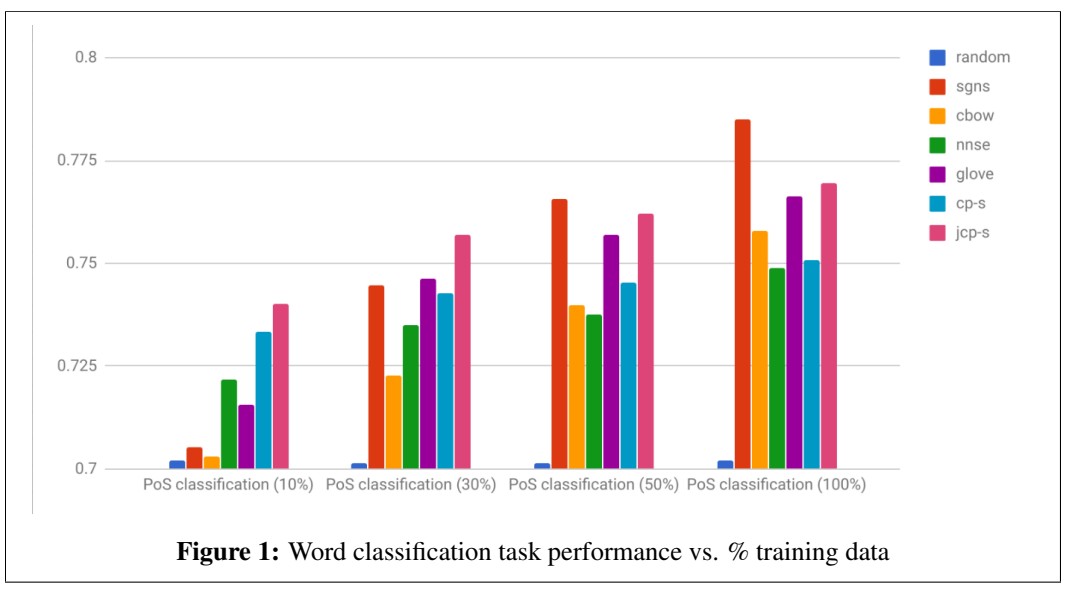

**Figure 1:** Word classification task performance vs. % training data

At this task, we notice that when 100% of the training data is presented, SGNS outperforms all other embeddings at this task. Since this task is a measure of quality of syntactic information encoded in an embedding, these results are an indication that perhaps SGNS encodes the most amount of syntactic information of all the embeddings presented here.

However, when training data is limited (restricting training data to 10%/30%), it is evident that our tensor-based embeddings outperform the other baselines. So even though SGNS eventually is able to provide higher quality syntactic information when enough training data is provided, this information is encoded more readily in our embeddings for supervised tasks to use, a favorable property for the sake of transfer learning.

**Table 3:** Supervised sentiment analysis scores across all embeddings

| (Method) | 10% training data | 30% training data | 50% training data | 100% training data |
|:---:|:---:|:---:|:---:|:---:|
| **Random** | 0.6999 | 0.7254 | 0.7311 | 0.7337 |
| **SGNS** | 0.7348 | 0.7590 | 0.7643 | 0.7696 |
| **CBOW** | 0.7322 | 0.7537 | 0.7591 | 0.7644 |
| **NNSE** | 0.7233 | 0.7476 | 0.7531 | 0.7572 |
| **GloVe** | 0.7310 | 0.7564 | 0.7622 | 0.7670 |
| **CP-S** | 0.7214 | 0.7454 | 0.7514 | 0.7575 |
| **JCP-S** | **0.7460** | **0.7681** | **0.7732** | **0.7774** |

**Sentiment analysis results.** The results are shown in **Table 3**. In this task, JCP-S is the dominant method across all levels of training data, further showing that exploiting both second and third order co-occurrence data leads to higher quality semantic information being encoded in the embedding.

Further theoretical research is needed to understand why JCP-S outperforms CP-S at this task, but its superior performance relative to the baselines across all levels of training data in these experiments demonstrates the quality of semantic information encoded by JCP-S.

Based on the clear relative success of our embeddings on these tasks, we can see that utilizing tensor factorization to create word representations has the propensity to encode a greater amount of semantic information than standard state-of-the-art pairwise or matrix-based word embedding methods.

**Word Similarity results.**

**Table 4:** Word Similarity Scores (Spearman's $\rho$)

| (Method) | MEN | MTurk | RW | SimLex999 |
|:---:|:---:|:---:|:---:|:---:|
| **Random** | 0.04147 | -0.0382 | -0.0117 | 0.0053 |
| **SGNS** | 0.5909 | 0.5342 | **0.3704** | 0.2264 |
| **CBOW** | 0.5537 | 0.4225 | 0.3444 | **0.2727** |
| **NNSE** | 0.5055 | 0.5068 | 0.1993 | 0.1263 |
| **GloVe** | 0.4914 | 0.4733 | 0.1750 | 0.1403 |
| **CP-S** | 0.4723 | 0.4738 | 0.0875 | 0.0399 |
| **JCP-S** | **0.6158** | **0.5343** | 0.3546 | 0.2272 |

We show the word similarity results in **Table 4**. As we can see, our embeddings perform competitively with the state-of-the-art at these tasks. It is worth including these results as the word similarity task is a very common way of evaluating embedding quality in the literature. However, due to the many intrinsic problems with evaluating word embeddings using word similarity (Faruqui et al. (2016)), we do not read further into these results.

## MULTIPLICATIVE COMPOSITIONALITY

We find that even though they are not explicitly trained to do so, our tensor-based embeddings capture polysemy information naturally through multiplicative compositionality. We demonstrate this property qualitatively and provide proper motivation for it, leaving automated utilization to future work.

In our tensor-based embeddings, we found that one can create a vector that represents a word $w$ in the context of another word $w'$ by taking the elementwise product $\mathbf{v}_w * \mathbf{v}_{w'}$. We call $\mathbf{v}_w * \mathbf{v}_{w'}$ a "meaning vector" for the polysemous word $w$.

For example, consider the word *star*, which can denote a lead performer or a celestial body. We can create a vector for *star* in the "lead performer" sense by taking the elementwise product $\mathbf{v}_{star} * \mathbf{v}_{actor}$. This produces a vector that lies near vectors for words related to lead performers and far from those related to *star*'s other senses.

**Table 5:** Nearest neighbors (in cosine similarity) to elementwise products of word vectors

| Composition | Nearest neighbors (CP-S) | Nearest neighbors (JCP-S) | Nearest neighbors (CBOW) |
|---|---|---|---|
| *star* $*$ *actor* | *oscar*, *award-winning*, *supporting* | *roles*, *drama*, *musical* | *DNA*, *younger*, *tip* |
| *star* $+$ *actor* | *stars*, *movie*, *actress* | *actress*, *trek*, *picture* | *actress*, *comedian*, *starred* |
| *star* $*$ *planet* | *planets*, *constellation*, *trek* | *galaxy*, *earth*, *minor* | *fingers*, *layer*, *arm* |
| *star* $+$ *planet* | *sun*, *earth*, *galaxy* | *galaxy*, *dwarf*, *constellation* | *galaxy*, *planets*, *earth* |
| *tank* $*$ *fuel* | *liquid*, *injection*, *tanks* | *vehicles*, *motors*, *vehicle* | *armored*, *tanks*, *armoured* |
| *tank* $+$ *fuel* | *tanks*, *engines*, *injection* | *vehicles*, *tanks*, *powered* | *tanks*, *engine*, *diesel* |
| *tank* $*$ *weapon* | *gun*, *ammunition*, *tanks* | *brigade*, *cavalry*, *battalion* | *persian*, *age*, *rapid* |
| *tank* $+$ *weapon* | *tanks*, *armor*, *rifle* | *tanks*, *battery*, *batteries* | *tanks*, *cannon*, *armored* |

To motivate why this works, recall that the values in a third order PPMI tensor $\mathcal{M}$ are given by:

$$m_{ijk} = PPMI(w_i, w_j, w_k) \approx \sum_{r=1}^{R} v_{ir} v_{jr} v_{kr} = \langle \mathbf{v}_i * \mathbf{v}_j, \mathbf{v}_k \rangle,$$

where $\mathbf{v}_i$ is the word vector for $w_i$. If words $w_i, w_j, w_k$ have a high PPMI, then $\langle \mathbf{v}_i * \mathbf{v}_j, \mathbf{v}_k \rangle$ will also be high, meaning $\mathbf{v}_i * \mathbf{v}_j$ will be close to $\mathbf{v}_k$ in the vector space by cosine similarity.

For example, even though *galaxy* is likely to appear in the context of the word *star* in in the "celestial body" sense, $\langle \mathbf{v}_{star} * \mathbf{v}_{actor}, \mathbf{v}_{galaxy} \rangle \approx$ PPMI(*star, actor, galaxy*) is low whereas $\langle \mathbf{v}_{star} * \mathbf{v}_{actor}, \mathbf{v}_{drama} \rangle \approx$ PPMI(*star, actor, drama*) is high. Thus , $\mathbf{v}_{star} * \mathbf{v}_{actor}$ represents the meaning of *star* in the "lead performer" sense.

In **Table 5** we present the nearest neighbors of multiplicative and additive composed vectors for a variety of polysemous words. As we can see, the words corresponding to the nearest neighbors of the composed vectors for our tensor methods are semantically related to the intended sense *both* for multiplicative and additive composition. In contrast, for CBOW, only additive composition yields vectors whose nearest neighbors are semantically related to the intended sense. Thus, our embeddings can produce complementary sets of polysemous word representations that are qualitatively valid whereas CBOW (seemingly) only guarantees meaningful additive compositionality.

While we leave automated usage and measuring of this property to future work, it is interesting to see that this property is empirically validated in our embeddings and not the existing baselines after motivating its existence from a tensor factorization point of view.

## CONCLUSION

Our key contributions are as follows:

1. **Two novel tensor factorization based word embeddings.** We presented CP-S and JCP-S, which are word embedding techniques based on symmetric CP decomposition. We experimentally demonstrated that these embeddings outperform existing state-of-the-art matrix-based techniques on a number of downstream tasks when trained on the same data.

2. **A novel joint symmetric tensor factorization problem.** We introduced and utilized Joint Symmetric Rank-$R$ CP Decomposition to train JCP-S. In this problem, multiple supersymmetric tensors must be decomposed using a *single* rank-$R$ factor matrix. This technique allows for utilization of both second and third order co-occurrence information in word embedding training.

3. **A new embedding evaluation metric to measure amount of third order information.** We produce a 3-way analogue of Outlier Detection (Camacho-Collados & Navigli (2016)) that we call OD3. This metric evaluates the degree to which third order information is captured by a given word embedding. We demonstrated this by showing our tensor based techniques, which naturally encode third information, perform considerably better at OD3 compared to existing second order models.

4. **Word vector multiplicative compositionality for polysemous word representation.** We showed that our word vectors can be meaningfully composed *multiplicatively* to create a "meaning vector" for each different sense of a polysemous word. This property is a consequence of the higher order information used to train our embeddings, and was empirically shown to be unique to our tensor-based embeddings.

Tensor factorization appears to be a highly applicable and effective tool for learning word embeddings, with many areas of potential future work. Leveraging higher order data in training word embeddings is useful for encoding new types of information and semantic relationships compared to models that are trained using only pairwise data. This indicates that such techniques are useful for training word embeddings to be used in downstream NLP tasks.

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
