# OpenReview forum: "LEARNING SEMANTIC WORD RESPRESENTATIONS VIA TENSOR FACTORIZATION"
_ICLR.cc/2018/Conference — Reject_

### Official Review · AnonReviewer3 · 2017-11-21
**Good start, but more analysis and experiments needed.**

**Rating:** 5
**Confidence:** 3

**Review:**

In this paper, the authors consider symmetric (3rd order) CP decomposition of a PPMI tensor M (from neighboring triplets), which they call CP-S. Additionally, they propose an extension JCP-S, for n-order tensor decompositions. This is then compared with random, word2vec, and NNSE, the latter of two which are matrix factorization based (or interpretable) methods. The method is shown to be superior in tasks of 3-way outlier detection, supervised analogy recovery, and sentiment analysis. Additionally, it is evaluated over the MEN and Mturk datasets.


For the JCP-S model, the loss function is unclear to me. L is defined for 3rd order tensors only;  how is the extended to n > 3? Intuitively it seems that L is redefined, and for, say, n = 4, the model is M(i,j,k,n) = \sum_1^R u_ir u_jr u_kr u_nr. However, the statement "since we are using at most third order tensors in this work" I am further confused. Is it just that JCP-S also incorporates 2nd order embeddings? I believe this requires clarification in the manuscript itself.

For the evaluations, there are no other tensor-based methods evaluated, although there exist several well-known tensor-based word embedding models existing:

Pengfei Liu, Xipeng Qiu∗ and Xuanjing Huang, Learning Context-Sensitive Word Embeddings with Neural Tensor Skip-Gram Model,  IJCAI 2015

Jingwei Zhang and Jeremy Salwen, Michael Glass and Alfio Gliozzo. Word Semantic Representations using Bayesian Probabilistic Tensor Factorization, EMNLP 2014

Mo Yu, Mark Dredze, Raman Arora, Matthew R. Gormley, Embedding Lexical Features via Low-Rank Tensors

to name a few via quick googling.

Additionally, since it seems the main benefit of using a tensor-based method is that you can use 3rd order cooccurance information, multisense embedding methods should also be evaluated. There are many such methods, see for example

Jiwei Li, Dan Jurafsky, Do Multi-Sense Embeddings Improve Natural Language Understanding?

and citations within, plus quick googling for more recent works.

I am not saying that these works are equivalent to what the authors are doing, or that there is no novelty, but the evaluations seem extremely unfair to only compare against matrix factorization techniques, when in fact many higher order extensions have been proposed and evaluated, and especially so on the tasks proposed (in particular the 3-way outlier detection).

Observe also that in table 2, NNSE gets the highest performance in both MEN and MTurk. Frankly this is not very surprising; matrix factorization is very powerful, and these simple word similarity tasks are well-suited for matrix factorization. So, statements like "as we can see, our embeddings very clearly outperform the random embedding at this task" is  an unnecessary inflation of a result that 1) is not good and 2) is reasonable to not be good.

Overall, I think for a more sincere evaluation, the authors need to better pick tasks that clearly exploit 3-way information and compare against other methods proposed to do the same.

The multiplicative relation analysis is interesting, but at this point it is not clear to me why multiplicative is better than additive in either performance or in giving meaningful interpretations of the model.

In conclusion, because the novelty is also not that big (CP decomposition for word embeddings is a very natural idea) I believe the evaluation and analysis must be significantly strengthened for acceptance.

---

> ### Author Response · Authors · 2017-12-18
> **reply to reviewer 3**
>
> The reviewer's comments are marked with a single * and our responses are marked with double **
>
> * "Is it just that JCP-S also incorporates 2nd order embeddings?"
> ** Exactly correct. We apologize for the confusing wording, and will update it in revised versions to be more clear. JCP-S is simultaneously decomposing a second and third order tensor using a single factor matrix.
>
> * No tensor-based baselines
> ** Let's go over the mentioned tensor-based approaches (which we also considered when formulating the idea for this paper):
> "Learning Context-Sensitive Word Embeddings with Neural Tensor Skip-Gram Model" - Their best word embedding by far results from making the tensor's third axis... one dimensional? So they are factoring a |V| x |V| x 1 "tensor"? This is really just matrix factorization.
> "Word Semantic Representations using Bayesian Probabilistic Tensor Factorization" doesn't apply because it uses supervised data, and we are considering unsupervised pre-trained embeddings. While they use the CP decomposition, they do not consider a symmetric decomposition based on an unsupervised corpus, which is the problem we are considering.
> "Embedding Lexical Features via Low-Rank Tensors" - While they do use word embeddings and CP decomposition, they are not pre-training generic word embeddings in this paper and thus we cannot use their methods to compare against ours.
> "Explaining and Generalizing Skip-Gram through Exponential Family Principal Component Analysis" is focused on creating an embedding to capture syntactic information, whereas ours are more semantically-focused and as such we evaluate our embeddings on a bed of semantic tasks, a testbed on which it would be unfair to compare a syntactic-focused embedding.
> ** Thus, the pointed tensor-based approaches to creating word embeddings are relatively orthogonal to our ideas.
>
> * Word similarity result wording
> ** We agree that matrix factorization is well-suited to modeling word similarity, and likely optimal in some sense for the task. However, with our comment, we are trying to emphasize the fact that we are including this result for completeness and not taking it heavily. We apologize if the wording sounds arrogant and will change the wording in future iterations of the paper.
>
> Again, please keep an eye out for a revised version of the paper, which will include further baselines, in particular, other state-of-the-art embedding techniques, which all reviewers seemed to agree that our paper needed. We hope that such updated results will alleviate some of your concerns about the evaluation of our techniques, and further convince you of the utility of our embeddings.

---

> > ### Comment · AnonReviewer3 · 2018-01-07
> > **Response to rebuttal and revision**
> >
> > After reading the revision I cannot change my rating. My biggest issue is still the lack of comparison against a tensor-based, or at least a multisense embedding. The papers I listed were after some quick googling and I am still unclear as to why they are unacceptable comparisons (except perhaps the second one). If you can show your work outperforms these other works, agreed that it’s not apples to apples but it’s still much closer than comparing a tensor based approach against a slew of matrix based approaches, of which one can never expect promising behavior on the contextual tasks you listed.

---

> > > ### Author Response · Authors · 2018-01-17
> > > **Tensor-based embedding results**
> > >
> > > Hi Reviewer 3,
> > >
> > > We see your point about needing to compare against another tensor method. While we are unaware of another embedding method that directly compares to ours (semantically-focused tensor factorization-based embedding), we see the utility in comparing against another embedding method that utilizes tensor factorization.
> > > So, we implemented HOSG from the fourth paper you mentioned as it was the closest to our approach -- still an unsupervised word embedding. We used the best method found in the paper -- context window of 5 and using third order factorization of the positional third order tensor as it performed better than SG more consistently in their experiments, using the recommended hyperparameters and the same dataset we used in the original paper for a fair comparison. The results on all tasks are linked below in an anonymous data dump, including the results of our embeddings and baselines for easy comparison.
> > >
> > > https://pastebin.com/Skyfa44v
> > >
> > > The results are as we expected -- it outperforms both our embeddings and the baselines at the synactic-based PoS task when the greatest amount of training data is presented to the supervised model (HOSG also performs best at the MTurk WS dataset). This makes sense since it was motivated in the original paper that this embedding would be more focused on encoding syntactic information. Still, we see our embeddings continue to outperform the baseline even at the synactic task when supervised data is poor (indicating information accessibility) and at semantically-focused tasks like OD(n) and Sentiment Analysis.
> > >
> > > It is also worth noting that it makes sense that HOSG does not outperform SG on OD3 since it is not actually being trained using information about groups of 3 words (which is what OD3 is testing for), but rather using the third tensor dimension to augment the syntactic positions in the context matrix.
> > >
> > > We can of course include these results in the final version of the paper if need be. Let us know if these results change your stance on the paper at all. Also, if other reviewers were concerned about other tensor-based embeddings, please consider the discussion in this comment.
> > >
> > > Thank you.

---

### Official Review · AnonReviewer2 · 2017-11-26
**A tensor extension of the familiar PPMI factorization for learning word embeddings**

**Rating:** 5
**Confidence:** 5

**Review:**

The paper proposes to extend the usual PPMI matrix factorization (Levy and Goldberg, 2014) to a (3rd-order) PPMI tensor factorization. The paper chooses symmetric CP decomposition so that word representations are tied across all three views. The MSE objective (optionally interpolated with a 2nd-order tensor) is optimized incrementally by SGD.

The paper's most clear contribution is the observation that the objective results in multiplicative compositionality of vectors, which indeed does not seem to hold in CBOW.

While the paper reports superior performance, the empirical claims are not well substantiated. It is *not* true that given CBOW, it's not important to compare with SGNS and GloVe. In fact, in certain cases such as unsupervised word analogy, SGNS is clearly and vastly superior to other techniques (Stratos et al., 2015). The word similarity scores are also generally low: it's easy to achieve >0.76 on MEN using the plain PPMI matrix factorization on Wikipedia. So it's hard to tell if it's real improvement.

Quality: Borderline. The proposed approach is simple and has an appealing compositional feature, but the work is not adequately validated and the novelty is somewhat limited.

Clarity: Clear.

Originality: Low-rank tensors have been used to derive features in many prior works in NLP (e.g., Lei et al., 2014). The paper's particular application to learning word embeddings (PPMI factorization), however, is new although perhaps not particularly original. The observation on multiplicative compositionality is the main strength of the paper.

Significance: Moderate. For those interested in word embeddings, this work suggests an alternative training technique, but it has some issues (described above).

---

> ### Author Response · Authors · 2017-12-18
> **reply to reviewer 2**
>
> The reviewer's comments are marked with a single * and our responses are marked with double **
>
> * "It is *not* true that given CBOW, it's not important to compare with SGNS and GloVe. In fact, in certain cases such as unsupervised word analogy, SGNS is clearly and vastly superior to other techniques (Stratos et al., 2015)."
> ** Thank you for pointing this out. We are training these embeddings on the same data the other embeddings were trained on using the recommended hyperparameters, and will include both SGNS and GloVe as baselines in the updated version of the paper.
>
> * "The word similarity scores are also generally low: it's easy to achieve >0.76 on MEN using the plain PPMI matrix factorization on Wikipedia. So it's hard to tell if it's real improvement"
> ** We repeat the statement from our paper that word similarity is a poor metric for evaluating the quality of information held in a word embedding, and performance at word embedding is poorly correlated with performance at downstream tasks. As such, we are less worried about performing well at the word similarity task compared to the other tasks shown to be more relevant to the practical use of word embeddings.
> ** Also, remember that we are providing a preliminary exploration of this approach, and thus only use a much smaller dataset (but still non-trivial - ~150M tokens) than production-ready approaches such as pre-trained GloVe or word2vec, and thus do not expect our numbers to be directly comparable to other published approaches.
> To reviewer 2: Please keep an eye out for revised versions of our paper after we have time to consider more experiments. Hopefully, our next form of evaluation will be more compelling after we update with even more of the common baselines.

---

### Official Review · AnonReviewer1 · 2017-11-27
**The paper approaches the important problem of word embeddings via factorization of a pointwise mutual information tensor. The idea is not novel and the proposed algorithm lacks theoretical guarantees, despite this is one of the main motivations. Although extensive experimental evaluation with good results is provided, it is hard to assess due to the lack of details about the experiments and missing comparisons with other important algorithms.**

**Rating:** 5
**Confidence:** 5

**Review:**

The paper presents the word embedding technique which consists of: (a) construction of a positive (i.e. with truncated negative values) pointwise mutual information order-3 tensor for triples of words in a sentence and (b) symmetric tensor CP factorization of this tensor. The authors propose the CP-S (stands for symmetric CP decomposition) approach which tackles such factorization in a "batch" manner by considering small random subsets of the original tensor. They also consider the JCP-S approach, where the ALS (alternating least squares) objective is represented as the joint objective of the matrix and order-3 tensor ALS objectives. The approach is evaluated experimentally on several tasks such as outlier detection, supervised analogy recovery, and sentiment analysis tasks.

CLARITY: The paper is very well written and is easy to follow. However, some implementation details are missing, which makes it difficult to assess the quality of the experimental results.

QUALITY: I understand that the main emphasis of this work is on developing faster computational algorithms, which would handle large scale problems, for factorizing this tensor. However, I have several concerns about the algorithms proposed in this paper:

  - First of all, I do not see why using small random subsets of the original tensor would give a desirable factorization. Indeed, a CP decomposition of a tensor can not be reconstructed from CP decompositions of its subtensors. Note that there is a difference between batch methods in stochastic optimization where batches are composed of a subset of observations (which then leads to an approximation of desirable quantities, e.g. the gradient, in expectation) and the current approach where subtensors are considered as batches. I would expect some further elaboration of this question in the paper. Although similar methods appeared in the tensor literature before, I don't see any theoretical ground for their correctness.

  - Second, there is a significant difference between the symmetric CP tensor decomposition and the non-negative symmetric CP tensor decomposition. In particular, the latter problem is well posed and has good properties (see, e.g., Lim, Comon. Nonengative approximations of nonnegative tensors (2009)). However, this is not the case for the former (see, e.g., Comon et al., 2008 as cited in this paper). Therefore, (a) computing the symmetric and not non-negative symmetric decomposition does not give any good theoretical guarantees (while achieving such guarantees seems to be one of the motivations of this paper) and (b) although the tensor is non-negative, its symmetric factorization is not guaranteed to be non-negative and further elaboration of this issue seem to be important to me.

  - Third, the authors claim that one of their goals is an experimental exploration of tensor factorization approaches with provable guarantees applied to the word embedding problem. This is an important question that has not been addressed in the literature and is clearly a pro of the paper. However, it seems to me that this goal is not fully implemented. Indeed, (a) I mentioned in the previous paragraph the issues with the symmetric CP decomposition and (b) although the paper is motivated by the recent algorithm proposed by Sharan&Valiant (2017), the algorithms proposed in this paper are not based on this or other known algorithms with theoretical guarantees. This is therefore confusing and I would be interested in the author's point of view to this issue.

  - Further, the proposed joint approach, where the second and third order information are combined requires further analysis. Indeed, in the current formulation the objective is completely dominated by the order-3 tensor factor, because it contributes O(d^3) terms to the objective vs O(d^2) terms contributed by the matrix part. It would be interesting to see further elaboration of the pros and cons of such problem formulation.

  - Minor comment. In the shifted PMI section, the authors mention the parameter alpha and set specific values of this parameter based on experiments. However, I don't think that enough information is provided, because, given the author's approach, the value of this parameter most probably depends on other parameters, such as the bach size.

  - Finally, although the empirical evaluation is quite extensive and outperforms the state-of the art, I think it would be important to compare the proposed algorithm to other tensor factorization approaches mentioned above.

ORIGINALITY: The idea of using a pointwise mutual information tensor for word embeddings is not new, but the authors fairly cite all the relevant literature. My understanding is that the main novelty is the proposed tensor factorization algorithm and extensive experimental evaluation. However, such batch approaches for tensor factorization are not new and I am quite skeptical about their correctness (see above). The experimental evaluation presents indeed interesting results. However, I think it would also be important to compare to other tensor factorization approaches. I would also be quite interested to see the performance of the proposed algorithm for different values of parameters (such as the butch size).

SIGNIFICANCE: I think the paper addresses very interesting problem and significant amount of work is done towards the evaluation, but there are some further important questions that should be answered before the paper can be published.  To summarize, the following are the pros of the paper:

  - clarity and good presentation;
  - good overview of the related literature;
  - extensive experimental comparison and good experimental results.

While the following are the cons:

  - the mentioned issues with the proposed algorithm, which in particular does not have any theoretical guarantees;
  - lack of details on how experimental results were obtained, in particular, lack of the details on the values of the free parameters in the proposed algorithm;
  - lack of comparison to other tensor approaches to the word embedding problem (i.e. other algorithms for the tensor decomposition subproblem);
  - the novelty of the approach is somewhat limited, although the idea of the extensive experimental comparison is good.

---

> ### Author Response · Authors · 2017-12-18
> **Reply to reviewer 1**
>
> Comments by reviewer start with a single * and our responses start with double star **
> * "The main emphasis of this work is ....."
> ** Actually, the main emphasis of our work is exploring the utility of considering higher order approaches than just pairs of words. We make few arguments based on the computational superiority of our work.
>
> ** JCP-S has nothing to do with ALS -- this is a misconception by the reviewer.
> * "I do not see why using small random subsets ...."
> ** See "Expected Tensor Factorization with Stochastic Gradient Descent" by T Maehara (2016). We consider the same problem.We believe the reviewer misunderstood the objective we are actually minimizing. If we consider the entire PPMI tensor to be the full "set of observations", then we are exactly taking subsets of the full set of observations. Explicitly, if we represent the entire tensor as a list of 4-tuples of the indices and values of the nonzero entries L := [(x1, y1, z1, val1), (x2, y2, z2, val2), ..., (xnnz, ynnz, znnz, valnnz)], then each batch i we consider is a strict subset Li ⊊ L where ∪i Li = L. Because of this, we actually are considering the batch stochastic optimization setting that the reviewer was talking about.
> ** We also do not attempt to prove any theoretical guarantees for why this approach works. We simply demonstrate empirically that it improves the quality of certain types of information encoded for specific classes of NLP tasks, and are motivated by the many applications of tensor decompositions to existing ML problems.
>
> * Symmetric and/or non-negative CP Decomposition and related tensor literature.
> ** The reviewer’s points are well-taken. We realize that our coverage of the relevant papers on theoretical properties and results on symmetric tensor factorization was rather incomplete and could have been misleading to the reader. We will revise it adequately. In Comon et. al’s 2008 paper and see also https://web.stanford.edu/group/mmds/slides/lim-mmds.pdf - slide 26, the main result states symmetric CP decomp of a symmetric always exists over complex numbers, doesn’t matter if the tensor is non-negative or not. Further (see proposition 5.3 in Comon et al.’s 2008 paper) it also follows that when order < dim, then symmetric CP rank = CP rank generically. While our tensor is real and we are optimizing over the real space, in this case the symmetric CP rank can exceed the dimension (in particular, can be larger than \binom{k + n -1}{n-1}, k=3, n=size of vocab), nevertheless, as is done in many application papers that factor the third order cumulant, we are merely seeking *a* real symmetric rank-R approximation (may not be the best) to the symmetric tensor.
> ** We actually did spend a good amount of time considering the non-negative symmetric CP decomposition, and called it CP-SN. We also tried a non-negative joint symmetric CP decomposition we called JCP-SN. However, in all of the experiments we tried, the performance of these non-negative embeddings never surpassed that of the unconstrained varieties we presented in this paper, so we decided to omit it from the paper in favor of clarity and conciseness. However, we concede that it perhaps would have been smart to acknowledge such experiments in light of the nice theoretical properties of non-negative CP decompositions that you mentioned, although it can still be computationally hard to obtain).
>
> * JCP-S being dominated by the third order tensor
> ** Again, we provide no theoretical properties or guarantees of our methods. We agree that such research would likely be illuminating and would probably lead to gains and extensions to our approaches, but at this point we are merely providing preliminary empirical results to a new way to encode word information in vectors.
>
> * "I think it would be important to compare ....."
> ** Regarding Sharan and Valiant [2017], we are merely pointing out that in contrast to the results in that paper, we show that tensor factorization of 3-way co-occurrence can outperform the matrix based approaches on a number of different tasks -- our research was conducted at the same time as theirs, and was in fact not inspired by their work at all.
> ** In light of this discussion, it is worth noting that our paper is primarily aimed to demonstrate the utility of symmetric tensor factorization and encoding higher order information for word embeddings, rather than claim to be the current "state-of-the-art" in word embeddings (if such a thing were to exist). If we were to claim the latter, we would have to train on a much larger dataset (production-scale), but we instead aim to show that our methods can be used to encode certain types of information better than those that do not take higher-order co-occurrence information into account.
> We are considering them heavily for the revised version of our paper (to be uploaded shortly).

---

### Author Response · Authors · 2017-12-30
**Notes on the revised version of the paper**

Hi all,
We have uploaded the edited version of our paper. Based on your feedback, we included more detail in hope of clarify the utility of our work.
We have now adequately revised related tensor factorization literature and expanded the related work section. On the evaluation front, the major changes to the paper include:

1) We replaced the neural network task with supervised part of speech classification based on the word's embedding. The reason we did this was because we noticed a troubling trend when fine-tuning the parameters of the network: the random embedding was achieving over 90% accuracy *on the test set* after sufficient training. This is higher than some of the best results we were getting for SGNS, GloVe, etc.
We believe this to be because of problems with the construction of the Google analogy dataset. There are 29 instances of "a : b :: Europe : Euro", so the neural network can simply learn the mapping Europe -> Euro, without even taking the words "a" and "b" into account. (Further, there are no instances of any "a : b :: Europe : d" for any d != Euro). We believe this is what the NN was doing for the random embedding, simply memorizing the word->word mapping without taking the rest of the analogy query into account.
Because the random embedding (which clearly encodes no information) was able to outperform well-established baselines, we no longer believe it should be used as a measure of the quality of information encoded in an embedding. Thus, rather than have just one fewer evaluation task, we decided to replace it with the different, simpler one of PoS classification.

2) Added SGNS and GloVe as baselines on the same dataset for a more robust comparison against the state-of-the-art techniques.

3) Re-trained *all* of our embeddings on a more recent dump of Wikipedia (mid-2017). Our previous results were trained on a 2008 dump of Wikipedia which we initially chose because it was readily available and already parsed. The more recent data should provide fresher/more recent word representations. We also decreased the minimum wordcount requirement from 2,000 times to 1,000 times, increasing the vocabulary count while still removing noisy words.

4) Included two more wordsim datasets (RW, SimLex999) to add more depth to the evaluation.

5) Included explicit hyperparameter settings we used and discussed how we found them.

Despite these rather large changes, the story of the evaluation remains fairly unchanged -- our embeddings continue to perform better at semantic tasks and in more data-sparse domains, indicating that they tend to encode more semantic information more readily, even when compared against state-of-the-art baselines including SGNS and GloVe.
We hope that these updated results are much more convincing than those in our original paper. We look forward to your feedback based on these updates! Thank you for your time and consideration.

---

### Decision · Program_Chairs · 2018-01-29
**ICLR 2018 Conference Acceptance Decision**

**Decision:**

Reject

**Comment:**

The reviewers are concerned that the evaluation quality is not sufficient to convince readers that the proposed embedding method is indeed superior to alternatives. Though the authors attempted to address these comments in a subsequent revision but still, e.g., the evaluation is only intrinsic or on contrived problems. Given the limited novelty of the approach (it is a fairly straightforward generalization of Levy and Goldberg's factorization of PPMI matrix; the factorization is not new per se as well), the quality of experiments and analysis should be improved.

+ the paper is well written
- novelty is moderate
- better evaluation and analysis are necessary